# A Study of the Behavior and Responsibility of Slovak Drivers, Especially in Case of Fatigue

Adrian Hajducik [1],*, Stefan Medvecky [2], Slavomir Hrcek [1] and Jaromir Klarak [3]

1 Department of Design and Machine Elements, Faculty of Mechanical Engineering, University of Žilina, 010 26 Žilina, Slovakia; slavomir.hrcek@fstroj.uniza.sk

2 Institute of Competitiveness and Innovation, University of Žilina, 010 26 Žilina, Slovakia; stefan.medvecky@fstroj.uniza.sk

3 Department of Automated Production Systems, University of Žilina, 010 26 Žilina, Slovakia; jaromir.klarak@fstroj.uniza.sk

* Correspondence: adrian.hajducik@fstroj.uniza.sk

**Abstract:** Driver fatigue can be manifested by various highly dangerous direct and indirect symptoms, for example, inattention or lack of concentration. The aim of the study was to compare the behavior of young drivers, older drivers and professional drivers, particularly in situations where they feel fatigued. In the online questionnaire, drivers answered various questions which analysed their responsibility of driving a car during fatigue, the optimum temperature in the car, or experience with microsleep. The sample of drivers consisted of 507 women and 951 men in Slovakia. Young drivers are more responsible when driving during fatigue, while professional drivers take risks, break the law, and drive tired more often. A total of 25% of all drivers experience fatigue more than once a week. Adverse results were found in connection with driving and fatigue, where more than 42% of respondents stated that their duties require them to drive even when they are tired. A total of 27% of drivers have had microsleep while driving. The survey showed that drivers are aware that thermoneutral temperature in a car interior can improve driving performance and a lower temperature can positively affect a person's attention. The regulation of the temperature in the car was helpful for 75% of all drivers when they felt tired, and more than 97% of the drivers lowered the temperature in the interior of the vehicle in order to achieve a better concentration. In addition to standard statistical methods, a neural network was used for the evaluation of the questionnaire, which sought for individual connections and subsequent explanations for the hypotheses. The applied neural network was able to determine parameters such as the age of the driver and the annual raid as the riskiest and closely associated with the occurrence of microsleep between drivers.

**Keywords:** responsible driving; Slovak drivers; interior temperature; fatigue; neural network





## 1. Introduction

The driving of a motor vehicle can be defined as a controlled activity of the driver, ensuring safe movement of the land vehicle such as a car, bus or truck. Driving skills are acquired through experience, age, predictability of collisions and so on. Thus, driving skills can be defined as a measure of the driver's ability to safely drive a vehicle. Driving skills are influenced by various environmental factors; thus, these skills themselves do not guarantee safety in traffic [1]. Driver fatigue/driver drowsiness is one of the most common causes of dangerous life-threatening situations [2], which studies have confirmed for several decades [3,4]. Driver fatigue is a topic that has been the subject of many scientific publications for decades [5,6]. Many studies point to possible causes of drowsiness or at-risk groups of drivers [7]. The causes of drowsiness are generally known today; this also applies to risk groups of drivers such as night shift workers, truck drivers [8] and so on [9,10]. The less serious consequences of drowsiness end up as material damage to

vehicles or public property. Damage to the health or lives of drivers, pedestrians and road users is generally considered to be a serious consequence of drowsiness.

A modern dangerous problem is also driver distraction caused by smartphones [11] or the complex digital interface of cars [12], causing a number of fatal accidents [13]. Mind-wandering also decreases the ability to focus on the driving task and is closely linked to the driver's fatigue or sleep hours [14]. Several studies have also addressed the driver's mental state [15], driver stress and emotions related to his behavior on the road [16]. In addition to the above, there are a number of other individual influences. We can therefore state that the driver of the car must be looked at comprehensively and from several perspectives [15].

A responsible driver can be defined as a driver who adheres to the speed limit, drives the vehicle and does not use a smartphone or various digital devices. Emotions and moods are not transferred to the driving style [11]. Other attributes are consideration, self-reflection and the ability not to take risks. In the context of this article and research, we associated the responsible driver primarily with feelings of fatigue and subsequent action. Additionally, on the basis of the presented studies, we state that one of the attributes of a responsible driver is the proper coping with feelings of fatigue and respect for fatigue as a dangerous phenomenon. A special case constitutes younger drivers who are considered to be less responsible in compliance with the traffic regulations [17–19]. There are several studies that consider this group as dangerous in terms of the occurrence of fatigue in this group. According to National Sleep Foundation's "Sleep in America" poll, only one in five adolescents (20%) receive an optimal amount of sleep during the week, and more than half (51%) report having driven drowsy in the past year [20–22]. Moreover, it is vital to ask how young drivers behave when fatigue occurs to evaluate their responsibility to drive during fatigue. One way to reduce the number of accidents caused by fatigue is to spread awareness of this phenomenon.

Slovakia is a transit country through which many international routes pass. A recent study on the occurrence of microsleep in drivers and their responsibility and attitude in the occurrence of this phenomenon was not conducted in Slovakia or in the surrounding countries (i.e., Poland, Czech, Hungary, Austria); this was one of the reasons for starting the research [23]. In addition, we believe, that it is very important to carry out more road safety research in countries that have traditionally received very little attention [24]. The article interprets knowledge about young, professional but also older drivers from Slovakia. This paper opens up space for the initiation of similar research in neighbouring countries, which can better explain road safety or traffic psychology in Europe. Moreover, there was a lack of public awareness of this phenomenon. This information will serve not only for the professional company but also for the relevant authorities in Slovakia, while informing the entire driving public.

Another part of the article analyses the temperature scales in the interior of the vehicle chosen by drivers during driving. At first glance, this seems to be taken out of context, but we have incorporated it into the questionnaire for the following reasons. At the same time, our team of experts is dedicated to monitoring the physiological parameters of the driver, such as ECG (electrocardiography), EMG (electromyography), GSR (galvanic skin response) and HRV (heart rate variability). When measuring the physiological GSR signal, there is not yet a complete consensus among researchers on whether this parameter is relevant in monitoring driver fatigue [25–27]. The basic problem associated with GSR is high sensitivity to atmospheric temperature [28,29]. Obviously, the self-report study is not the best way to analyse the effect of interior temperature to driver fatigue. However, the study highlighted temperature intervals, which will be considered as a starting point for ongoing research. In these temperature intervals, the relationship of GSR and driver fatigue will be further investigated.

The temperature of the vehicle interior has been addressed in several of the studies mentioned below. Basagaña et al. [30] addressed this issue due to the still unanswered question between interior temperature and possible driver fatigue. The study included 118,489 motor vehicle crashes. The study showed that the estimated risk of crashes with driver performance factors significantly increased by 1.1% for each 1 °C increase in maximum temperature. The authors of the study claim that motor vehicle crashes involving driver performance-associated factors were increased in association with heatwaves and increasing temperature. They consider these findings to be relevant for further studies and prevention plans.

The possible associations between climate parameters and driver fatigue have not been subject to specific studies thus far. That was the reason for starting the study for Makowiec-Dąbrowska et al. [31]. The study was conducted among 45 city bus drivers aged 31–58 years from Poland. The authors of the study claim that climate conditions can modify the drivers' fatigue; therefore, we should be aware of their impact on well-being. Shin et al. [32] and Magaña et al.'s [15] research also suggest a possible correlation and some effect of vehicle interior temperature on the driver's overall condition and attention.

There are many popular countermeasures that drivers take when they are tired, including stopping, getting out of the car, walking, and consuming coffee or other stimulating drinks [33–35]. Since the effect of ambient temperature on humans is described in this section and its effects are known thanks to the research presented, we were interested in the drivers' perception of ambient temperature when driving a motor vehicle.

Nowadays, there are several studies that are oriented towards the optimal temperature of the environment in which a person is located while performing a physical or mental activity [31]. The subject of these studies was the analysis of the ambient temperature for human activities. Initially, researchers focused on physical work and the impact of ambient temperature on performance during physical work or sports [36].

However, researchers in recent decades have examined the temperature of the environment and its impact on mentally working people, on their cognitive abilities, concentration, or the ability to learn at different environment temperatures [37]. There is a close correlation between human performance and the temperature of the immediate environment, as the research suggests. The human performance begins to decline significantly above 26 °C. There are not many studies of car temperature associated with the drivers' attention and drowsiness [32,38]. Nevertheless, the interior of a car is more specific and complex than a room or office. Ambient conditions such as the season and the intensity of sunlight or air quality have a great influence on the resulting indoor temperature [39,40]. Furthermore, based on the above-mentioned research, it can be concluded that the temperature of the interior of the vehicle should be between 18.5 °C and 27.5 °C and a lower temperature should be maintained for the comfort and concentration of the driver [41,42].

To sum up, the article answers questions on the following hypotheses.

**Hypothesis 1 (H1).** *Professional drivers feel tired more frequently than other drivers.*

**Hypothesis 2 (H2).** *The response to fatigue is related to the age of the driver.*

**Hypothesis 3 (H3).** *PD deal with fatigue different than other drivers.*

In addition, one of the research aims was to specify the ranges of temperatures chosen by drivers and to find out whether the lower temperature in the interior secures higher comfort and keeps drivers awake and, additionally, to determine the degree of importance that drivers put on the temperature inside the vehicle. In addition, it was investigated whether drivers assume a link between unfavourable interior temperature and driver fatigue.

## 2. Materials and Methods

The questionnaire was approved by an employee of the Bioethics Commission of the Ministry of Health of the Slovak Republic. The final questionnaire was distributed via the Internet and was directed and shared in communities and discussion forums where drivers and car fans gather. A direct link to the questionnaire was distributed along with its description in the relevant categories. In addition, it was an anonymous survey. We deliberately wanted more young drivers to fill in the questionnaire because we wanted a large enough sample. Apart from this, the questionnaire was distributed to a sample of professional drivers (PD) with the assistance of an organization, ČESMAD, in Slovakia, professionally oriented to road transport and considered to be the most influential associations in Slovakia. A total of 893 questionnaires were distributed via the Internet; 324 questionnaires were shared in communities. A total of 241 questionnaires were distributed with the assistance of an organization ČEMAD for truck drivers. The questionnaire was conducted over time (2 December 2019 to 2 February 2020). The questionnaire was supplemented by explanations. For example, a respondent was classified as a professional driver if they drove at least 30,000 km per year. Moreover, the frequency of car use had to be on a daily basis.

The questionnaire can be divided into 3 parts; Table 1 shows the questionnaire. The first part included information on the driver's gender, driver's age, annual mileage and frequency of use. The second part concerned fatigue, the incidence of fatigue while driving and the driver's subsequent action. Fatigue is one of the most common risk factors when driving because it causes drowsiness, decreases drivers' attention and may make them fall asleep at the wheel. In addition, this part of the questionnaire was supplemented by explanations and model situations that helped respondents to make better choices. The drivers also indicated whether they experienced microsleep while driving. Microsleep is a sleep state lasting from a few tens of milliseconds to half a minute. Microsleep is the result of fatigue and can cause dangerous situations with potentially fatal outcomes. The third part of the questionnaire aimed to find out the temperature values in which drivers feel comfortable and at the same time to confirm the recommended temperature interval in the interior of the vehicle. In another question, the drivers commented on and assessed the effect of temperature on their concentration. Concentration, generally, is the attention and focus of mental activity on a particular object, activity or plot and ignoring other disturbing influences.

**Table 1.** Framework of the questionnaire.

| $n = 1458$ | | |
|---|---|---|
| 1. Gender | | |
| Man | 65.3% | 952 |
| Woman | 34.7% | 507 |
| 2. Age | | |
| up to 25 years | 63% | 920 |
| 26–35 years | 23.4% | 342 |
| 36–50 years | 11.1% | 162 |
| 51–64 years | 2.3% | 34 |
| over 65 years | 0.2% | 4 |
| 3. Are you a professional driver? | | |
| Yes | 22.4% | 326 |
| No | 77.6% | 1131 |

**Table 1.** *Cont.*

| n = 1458 | | |
|---|---|---|
| **4. How many kilometers do you drive per year?** | | |
| Up to 5000 km | 29.8% | 435 |
| 5000–15,000 km | 25% | 365 |
| 15,000–30,000 km | 19.3% | 281 |
| 30,000–50,000 km | 10.6% | 154 |
| 50,000–90,000 km | 6.7% | 126 |
| 90,000 km and more | 8.6% | 98 |
| **5. How many days a week do you drive?** | | |
| Almost every day. | 53.8% | 784 |
| 3 to 5 times a week. | 20.6% | 374 |
| Less than 3 times. | 25.7% | 300 |
| **6. How often do you feel fatigued while driving a car?** | | |
| Almost every day. | 4% | 58 |
| More than 2 times a week. | 8.4% | 123 |
| Once a week. | 12% | 175 |
| 1 to 3 times a month. | 14% | 248 |
| Rarely than indicated. | 39.4% | 574 |
| Never. | 19.2% | 280 |
| **7. Do you drive during fatigue?** | | |
| I never drive during fatigue. | 13.7% | 200 |
| I avoid fatigue behind the wheel. | 40.8% | 595 |
| You cannot always avoid fatigue behind the steering wheel, in particular for duties and time schedule, therefore occasionally drive tired. | 42.3% | 617 |
| I often find myself tired behind the wheel. | 3.2% | 46 |
| **8. Have you already experienced microsleep while driving? (Microsleep is a sleep state lasting from a few tens of milliseconds to half a minute.)** | | |
| Yes | 27.9% | 406 |
| No | 72.1% | 1051 |
| **9. The appropriate temperature in the car's interior is largely a subjective value, it is recommended to keep the temperature in the range of 18–23 degrees Celsius. Do you think that the interior temperature outside the range of 18–23 °C affects the driver's attention? You may choose more than one option [30–32].** | | |
| The temperature inside the vehicle affects the driver's attention and fatigue. | 58.5% | 853 |
| I think that at a lower temperature a driver feels more comfortable and refreshed. | 34.4% | 501 |
| I think that higher temperatures in the interior of the vehicle can make a driver tired and cause adverse conditions such fatigue or loss of attention. | 49.7% | 725 |
| I think that only together with other factors such as (fatigue, sleep deprivation, night shift . . . ) inappropriate temperature (high or low) of the vehicle interior can adversely affect the driver's attention. | 44.3% | 646 |
| The temperature of the vehicle interior does not affect the driver. | 2.1% | 31 |
| I cannot answer the question. | 2.8% | 41 |

**Table 1.** *Cont.*

| n = 1458 | | |
|---|---|---|
| 10. At what interior temperature do you feel most comfortable. Choose from a range of temperatures. | | |
| less than 17 °C | 1.5% | 22 |
| 18–19 °C | 16.9% | 246 |
| 20–22 °C | 64.8% | 944 |
| 23–24 °C | 15.6% | 227 |
| 25 °C and above | 1.2% | 17 |
| 11. Did you help with the feelings of fatigue during driving, among other countermeasures by regulating the interior temperature? (It does not matter how the change was made (air conditioning, ventilation . . . ) | | |
| Yes | 73.3% | 1067 |
| No | 26.7% | 388 |
| 12. If so, in what way? | | |
| I tried to raise the interior temperature. | 2.3% | 26 |
| I tried to lower the interior temperature. | 97.7% | 1120 |

Table 3 applies to question 7, which was: "Do you drive during fatigue?". The degree of responsibility was divided into 4 responses. These answers were defined by the degree of responsibility and the individual answers were explained to the respondents in the questionnaire by a model situation. These responses combine attitudinal and behavioral measures. The attitudinal measure is the response: "It is not always possible to avoid fatigue behind the wheel, especially due to responsibilities and time schedule, so sometimes I drive tired.". The other three response options are behavioral measures.

The answer: "I never drive tired." indicates the highest degree of responsibility. This means that the driver does not break the law and does not endanger traffic and acts as required by law. So, when fatigue occurs, the respondent never drives and chooses to rest, stop and so on.

The answer: "I avoid fatigue behind the wheel." indicates the second degree of responsibility. It means a model situation in which the driver still drives when fatigue occurs, but uses all available countermeasures and, if this is not enough, the driver stops, takes a break, rests and so on. The difference between the first answer is mainly in the strictness of decision-making. In this case, the driver is more benevolent.

The answer: "It is not always possible to avoid fatigue behind the wheel, especially due to responsibilities and time schedule, so sometimes I drive tired." indicates a model situation, which already occurs on a regular basis, unlike the previous two. The driver engages in this situation repeatedly and slowly loses the need to take strong action against the onset of fatigue, because he begins to unconsciously accept fatigue and does not significantly suppress it. The driver is aware that fatigue is under control. However, we consider this to be dangerous behavior.

The answer: "I often find myself tired behind the wheel." is defined by a model situation wherein the driver loosely and benevolently approaches fatigue on a regular basis. The respondent drives even during fatigue and the driver's attempt to change this condition is not enough. This behavior is considered high risk and the driver violates the law and regulations on road traffic. Thanks to the model situations assigned to the individual answers, the respondents were able to distinguish the four options clearly.

### 2.1. Statistical Test Methods

Responses were downloaded in csv format. Incomplete questionnaires with missing data were completely excluded from the next process. The research methods used to

analyse the obtained data were, in addition to the tools of descriptive statistics, mainly the tools of inductive statistics. The data were evaluated using the IBM SPSS® v.25 (Armonk, New York, NY, USA). statistical package, analysing the variations with the Chi-square test, the Mann–Whitney test and Spearman's rho correlation. The Mann–Whitney test was used to work with the ordinal variable. The Mann–Whitney test answers the question of whether the difference between the medians of the two groups is statistically significant or only random. For example, the hypothesis that assumes that professional drivers feel tired more frequently than other drivers required the use of Mann–Whitney test. For the other hypotheses (Sections 3.2 and 3.3), two variables were used; in principle, it was an estimate of the expected number at appropriate numerical intervals. In addition, we worked with a nominal variable where it is appropriate to use the chi-square test.

### 2.2. Neural Network

The neural network was created in the Python 3.7 programming language using the Keras library (sequential model). The input layer was represented by 10 selected questions of the questionnaire. An experiment in the form of a neural network application consisted of creating a neural network that would be able to predict whether there is a link between responses from the questionnaire and question number 8. The aim was to examine which questions of the questionnaire are closely related to question number 8—a positive answer to this question. Simply put, if the respondent asked question 8 "YES", what were the other answers of the respondent? The principle was based on a random division of respondents into two groups. The first group was used for neural network training (training data) and the second (testing data), smaller group was subjected to evaluation and the search for answers closely linked to the affirmative answer to question 8.

The individual variables were confronted with real responses, which could theoretically suggest whether there was a possibility when the responses could determine their effect on microsleep in drivers. Weights—values of individual responses were determined based on the degree of influence of responses from individual layers of the neural network after training. As mentioned above, the created dataset was at a ratio of 80:20%, where 80% of the data was used as training data. The remaining 20% of the data was used for neural network validation. Training of the neural network was carried out in 1164 already filtered questionnaires. Learning parameters were defined for epochs = 2000 and batch size = 30. The ADAM optimization algorithm was chosen.

## 3. Results

A total of 1458 Slovak drivers completed the questionnaire. The sample of drivers consisted of 507 women and 951 men in Slovakia. The variables included gender, age of drivers, annual mileage and frequency of use of the car. The results shows that 77.4% of the sample were below 35 years of age. A total of 326 respondents were professional drivers (PD). For PD, in addition to the respondents' answers, we took into account an annual drive of at least 30,000 km per year and daily driving. These were the main conditions when we considered the driver to already be professional. The criterion was not whether driving was a job for the respondent. Table 1 shows the framework of the questionnaire. It was found that 30.3% of all drivers and 59.7% of PD experienced a dangerous state of microsleep. After analysis of the data, several claims were made.

### 3.1. PD Feel Tired More Frequently than Other Drivers

The following Table 2 applies to question 6, which was: "How often do you feel fatigued while driving a car?". It was found that 38.5% of drivers, regardless of whether they are PD or not, feel fatigued more than twice a week. As Table 2 shows, PDs are more dangerous in all responses. The hypothesis, which assumed that PD feel tired more frequently than other drivers, was confirmed. The hypothesis was verified using the Mann–Whitney test, comparing the frequency of fatigue in the mentioned groups (Table 2).

**Table 2.** Question 6—frequency of fatigue, answers with numerical values.

| | | | Professional Driver | | Total |
| | | | No | Yes | |
|---|---|---|---|---|---|
| Frequency of fatigue | 0—never | Count | 237 | 43 | 280 |
| | | % | 20.9% | 13.2% | 19.2% |
| | 1—Rarely than indicated | Count | 508 | 66 | 574 |
| | | % | 44.9% | 20.2% | 39.4% |
| | 2—Once to 3 times a month | Count | 189 | 59 | 248 |
| | | % | 16.7% | 18.1% | 17.0% |
| | 3—Once a week | Count | 120 | 55 | 175 |
| | | % | 10.6% | 16.9% | 12.0% |
| | 4—More than twice a week | Count | 55 | 68 | 123 |
| | | % | 4.9% | 20.9% | 8.5% |
| | 5—almost every day | Count | 23 | 35 | 58 |
| | | % | 2.0% | 10.7% | 4.0% |
| Total | | Count | 1132 | 326 | 1458 |
| | | % | 100.0% | 100.0% | 100.0% |

The level of statistical significance is <0.0005. The hypothesis was confirmed as the PDs feel fatigued significantly more frequently than other drivers. Spearman's rho-correlation also confirmed that the frequency of microsleep increases in direct proportion to the annual mileage.

*3.2. Responsibility When Fatigue Occurs and Age of the Driver*

After analysing the data, it is also clear from Table 3 that age and responsibility when fatigue occurs are related.

The hypothesis was verified by the Chi-square test of independence, where there are two variables in the relationship: the age of the driver and the attitudinal and behavioral measures when fatigue occurs.

**Table 3.** Question 7—Do you drive during fatigue? Answers x age of the driver.

| | | Age of the Driver | | | | Total |
| | | Up to 25 Years | 26–35 Years | 36–50 Years | 51+ Years | |
|---|---|---|---|---|---|---|
| I often find myself tired behind the wheel | Count | 25 | 14 | 6 | 1 | 46 |
| | % | 2.7% | 4.1% | 3.7% | 2.8% | 3.2% |
| It is not always possible to avoid fatigue behind the wheel, especially due to responsibilities and time schedule, so sometimes I drive tired | Count | 346 | 179 | 83 | 9 | 617 |
| | % | 37.7% | 52.3% | 51.2% | 25.0% | 42.3% |
| I never drive tired | Count | 144 | 29 | 18 | 9 | 200 |
| | % | 15.7% | 8.5% | 11.1% | 25.0% | 13.7% |
| I avoid fatigue behind the wheel | Count | 403 | 120 | 55 | 17 | 595 |
| | % | 43.9% | 35.1% | 34.0% | 47.2% | 40.8% |
| Total | Count | 918 | 342 | 162 | 36 | 1458 |
| | % | 100.0% | 100.0% | 100.0% | 100.0% | 100.0% |

The level of statistical significance (*p*-value) was 0.000. Hypothesis 2 was confirmed. The fatigue response behind the wheel is related to the driver's age.

### 3.3. Professional Drivers Deal with Fatigue Differently than Other Drivers

PDs are drivers who spend a lot of time on the road, having a lot of experience. However, the findings of how Slovak PDs behave when fatigue occurs are not exemplary and representative. Factors such as the mileage, frequency of car use and careless behavior when fatigue occurs multiply the risk by several times more than in casual drivers.

The hypothesis was verified by Chi-square tests of independence. There were two variables in the relationship, namely the attitude to fatigue and the answer of the driver (in this case, a PD). The level of statistical significance (*p*-value) was 0.000. Hypothesis 3 was confirmed.

A study of coping with fatigue among professional and other drivers showed in Table 4 the following results. Only 30.7% of PD avoid fatigue and only 11% of PD never drive during fatigue. PDs are expected to be responsible in complying with traffic rules. However, as can be seen from the table, 326 PD responded differently.

**Table 4.** Do you drive during fatigue? Attitude to fatigue among PD and other drivers.

| | | | Professional Driver | | Total |
|---|---|---|---|---|---|
| | | | No | Yes | |
| Attitude to fatigue | I often find myself tired behind the wheel | Count | 28 | 18 | 46 |
| | | % | 2.5% | 5.5% | 3.2% |
| | It is not always possible to avoid fatigue behind the wheel, especially due to responsibilities and time schedule, so sometimes I drive tired | Count | 445 | 172 | 617 |
| | | % | 39.3% | 52.8% | 42.3% |
| | I never drive tired | Count | 164 | 36 | 200 |
| | | % | 14.5% | 11.0% | 13.7% |
| | I avoid fatigue behind the wheel | Count | 495 | 100 | 595 |
| | | % | 43.7% | 30.7% | 40.8% |
| Total | | Count | 1132 | 326 | 1458 |
| | | % | 100.0% | 100.0% | 100.0% |

### 3.4. Interior Temperature

Study shows (Table 5) that 73.3% (1066) of all drivers regulate the interior temperature of the vehicle during fatigue, while 90% of these 73.3% drivers reduce the interior temperature to increase concentration and attention. This confirmed a relatively common phenomenon of lowering the temperature of the vehicle interior when drivers feel tired. Moreover, only 3% of all participating drivers assume that interior temperature does not affect driver fatigue.

**Table 5.** Car interior temperature regulation in case of driver fatigue and driver's age.

| | | | Age | | | | Total |
|---|---|---|---|---|---|---|---|
| | | | Up to 25 Years | 26–35 Years | 36–50 Years | 51+ Years | |
| Car interior temperature Regulation in case of driver fatigue | No | Count | 295 | 60 | 23 | 10 | 388 |
| | | % | 32.3% | 17.5% | 14.2% | 27.8% | 26.7% |
| | Yes | Count | 620 | 282 | 139 | 26 | 1066 |
| | | % | 67.8% | 82.5% | 85.8% | 72.2% | 73.3% |
| Total | | Count | 915 | 342 | 162 | 36 | 1454 |
| | | % | 100.0% | 100.0% | 100.0% | 100.0% | 100.0% |

Finally, 97% of all drivers consider the optimum temperature in the vehicle interior to be in the range of 18–24 °C. More specifically, (20–22 °C) 64% of drivers, (23–24 °C) 21% of drivers and (18–19 °C) 12% of drivers. Two percent of drivers reported the optimum temperature t > 25 °C and 1% of drivers reported t <17 °C as the ideal temperature.

Moreover, 95% of all respondents stated that temperature is a subjective and not a decisive factor in fatigue but with associated factors such as sleep deprivation, untreated sleep disorder or pre-trip fatigue can affect the driver.

### 3.5. An Experimental Approach in the Form of Neural Network Deployment

A wide neural network with five hidden layers was created. On the basis of the data obtained, this was a sufficient amount to obtain relevance to individual parameters in the learning process. The structure of the neural network is shown in Table 6.

**Table 6.** Structure of neural network.

| Layer | Inputs | Hidden | Hidden | Hidden | Hidden | Hidden | Output |
|---|---|---|---|---|---|---|---|
| Number | 10 | 10 | 100 | 100 | 5820 | 10 | 1 |

As illustrated in Figure 1, the neural network could achieve 90% training accuracy. Jumps in accuracy in the learning process are problematic due to the lack of data for training. For the sake of objectivity and validation, the neural network assessment is performed on 20% of the data not used in the training process. The overall evaluation of the trained neural network can be seen in Table 7.

**Table 7.** Overall evaluation of the trained neural network.

| Type of Data | Confidence > 50% | | | Confidence > 90% | | |
|---|---|---|---|---|---|---|
| | False | True | True% | False | True | True% |
| All | 105 | 1349 | 92.78 | 188 | 1266 | 87.07 |
| Training | 28 | 1136 | 97.59 | 64 | 1100 | 94.50 |
| Validation | 77 | 214 | 73.54 | 124 | 167 | 57.39 |

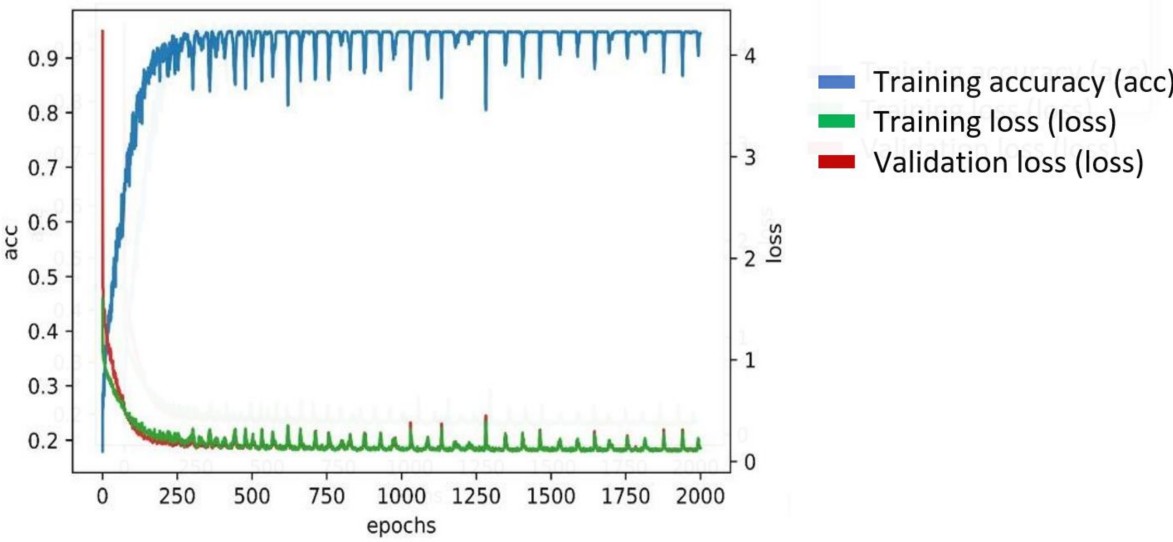

**Figure 1.** Neural network training process.

The degree of impact of individual responses on the occurrence of microsleep (positive answer to question 8) was determined based on the weights gained in the learning process. In the process of learning the neural network, the partial goal was to attempt to determine the weight, or the influence of individual responses and their theoretical connection to the previous experience with microsleep.

The second question concerned the age of the driver. The fourth question concerned the annual mileage. The neural network thus determined that these two questions were the most important when the respondent stated in the questionnaire their previous experience with microsleep. We consider the results generated by the neural network to be adequate and they create room for improvement and innovation in the design of the neural network for further purposes.

## 4. Discussion

This study provided an up-to-date view and data from a country that has traditionally received very little attention. This research was one of the first in Central Europe, which at least partially fill the gap, and bring several facts and conclusions, especially in case of the responsibility of drivers when fatigue occurs. Fatigue in drivers will most likely be impossible to avoid. On the other hand, we believe that even with the help of further research in the surrounding countries, notions such as consideration and responsibility will become known to the public. In addition to the influence of the driver's seniority and the driver's age, the research conducted by Jamroz [43] pointed to an important phenomenon, namely night driving. Moreover, this research confirmed the findings by Jamroz's research [43], that the riskiest group of drivers in terms of fatigue are drivers of productive age.

### 4.1. Age of the Driver and Response to Fatigue

Young drivers, frequently considered a risk group for violating rules and regulations [19], were not the most at risk in this study. The hypothesis that PD are more responsible on the road than young drivers has not been confirmed and has yielded rather worrying findings. Based on age, however, we can state that the riskiest or most irresponsible group in terms of attitudes to fatigue is the group of drivers aged 26–35 and 36–50 years. These groups achieve similar results. In them, irresponsibility can be mainly observed because they are exposed to probably the highest work and family load—responsibilities in working life, childcare and other family and related duties. The young drivers have shown a surprisingly positive and responsible approach. This phenomenon may be a positive outcome of the study, where we observe more responsible behavior in young drivers. However, the question is whether they will acquire this responsibility or lose it over time and thus approach the worse results.

A survey [44] that collected data from 19 European countries in 2015 identified as one of the main individual determinants of falling asleep younger age, especially male gender. It is interesting that in our research this phenomenon was not confirmed, despite the fact that we had a relatively large sample of young drivers. Likewise, the skill of the young drivers was surprising.

The incidence of micro-sleep among young drivers was only 16%. Drivers under 25 years, on the other hand, were surprisingly responsible in their attitudes to driving when they become tired. A total of 59.6% of young drivers do not drive during fatigue. This indicates that they follow the law and the principles of general health and road safety. On the other hand, a relatively low incidence of microsleep and driving while fatigued among young drivers could be attributed to the fact that young drivers simply have fewer opportunities to engage in such behaviors because of the relatively short amount of time they have been driving.

### 4.2. PD Manage with Fatigue Different than Other Drivers

The results of the proposed analysis confirmed the hypothesis that PD feel tired more frequently than other drivers. The alarming numbers indicate that 20.9% of PD feel tired more than twice a week and 11% of PD feel tired every day. Drivers are frequently overworked, which ultimately makes them the riskiest group of drivers. Moreover, 60% of PD have already experienced a micro-sleep, an extremely critical condition that frequently leads to traffic accidents. Moreover, more striking is the fact that only 41.7% of PD behave

according to the law and the driver regulations when fatigue occurs. This means that they stop driving the vehicle and take a break to restore physical strength and concentration. The remaining 58.3% of PD violate traffic rules and laws and drive tired for several reasons, e.g., hectic lifestyle, work responsibilities, schedule, etc. These findings have major public health and safety implications. Another research [45] focuses on professional drivers even more deeply and states that there is a significant dependence between professional driver fatigue and job seniority. Drivers with the longest seniority were the least likely to develop such symptoms as drowsiness, lack of concentration, irritability, or eye strain.

### 4.3. Interior Temperature

Research has also confirmed the claim that drivers are aware of the effect of temperature on driver concentration. Ninety-seven percent of all drivers, regardless of category, indicate the ideal temperature of the vehicle interior from 18 °C to 24 °C. Thanks to the answers, we can state that the drivers are aware of the influence of the interior temperature. Additionally, the driver assumes the existence of a link between unfavourable interior temperature and driver fatigue.

Our next task, as we mentioned in the introduction, will be to perform measurements of electrodermal activity (EDA) on a selected sample of people in this temperature range. We expect that the results should contribute to the issue between GSR (EDA) and driver fatigue. The obtained data will not only be a self-report study but will be verified on the basis of physiological measurements in these intervals. As interior temperature reduction has proved to be a widely used method in the survey, the next step will be to obtain experimental data on a sample of drivers. The true effectiveness of this countermeasure could be compared, for example, in terms of effect and time.

### 4.4. Neural Network

The last part of this article points out that it is possible to use a neural network not only to evaluate the questionnaire but also to search for connections that relate to the chosen subject of interest. The achieved accuracy of the neural network was 90%; as mentioned, this could be improved with a higher number of available data. It is clear that as the number of kilometers increases, the risk of microsleep increases. However, with the increasing age of drivers, it may be considered whether this risk increases with certainty. The output of the neural network was not new findings or theories. The output is a way for large-scale data to be evaluated. For a selected group of people, for example, young drivers, older drivers, motorcyclists and a defined survey parameter, this method seems to be an interesting procedure. Among other things, this research revealed shortcomings in the use of a neural network to evaluate questionnaires. We assume that the use of a neural network (NN) in the evaluation of the questionnaire is possible at least with the number of questionnaires of 5000 and more. We believe that with small numbers of questionnaires, NN is not able to interpret the results with an accuracy of more than 95%.

The neural network was partially fulfilled by now known facts, but in a less used way. However, the neural network reflected the answers of the respondents in this research. For a more general claim, it is necessary to use a larger amount of data. Finally, the use of a neural network in the search for connections has been used only on an experimental level and serves rather as a springboard for further research.

### 4.5. Future Research Directions and Limitations

The limits of this study include the number of analysed samples; the respondents were citizens of the Slovak Republic, and so this study represents a study of national scope. For future research, it would be interesting to include more European countries (Poland, Czech, Austria). Such a model could serve as an up-to-date view of the behavior of each group of drivers, because such an international study has been lacking in recent years in this region. This is also confirmed by the research of Schlembach et al. [46] which calls for research at national level, focusing on traffic safety culture.

We have several areas and improvements for future research in which we would like to continue and participate. Firstly, the description of the groups should be extended by other health data such as: BMI [47], health status, sleep problems such as obstructive sleep apnoea [48] and so on. These factors affect the overall condition of the driver and, of course, the adverse phenomenon itself, such as fatigue. We recommend incorporating no less important issues into the study in a suitable way, and that is night driving [2,48]. Moreover, this study focused more on age groups, but according to the following research, the influence of the driver's gender can play an important role [1,9], so it should not go unnoticed.

## 5. Conclusions

In summary, this study demonstrates the current state of drivers' behavior when they feel tired. In general, we can observe that 42.3% of drivers do not avoid a state of fatigue while driving, so we believe that drivers are frequently forced to drive by their circumstances, e.g., work schedule. Based on comparisons with other research, we can state that professional drivers and their fatigue are a constant problem not only on European roads. The research pointed to numbers that do not improve and rather fluctuate at similar values for several years. The article again emphasized and drew attention to increasing awareness, because it is one of the factors of how to fight this phenomenon.

One of the positive findings is the behavior of young drivers when they feel tired. However, we can attribute this to their young age and little experience. However, it is a fact on which it is possible to build in the future.

From the point of view of traffic safety, we consider this a high safety risk. In addition, this problem seems to have a persistent tendency. The results of a study that reflects the behavior of drivers have shown that this issue is not sufficiently solved and there is a lack of greater education, especially in countries such as Slovakia, which receive less attention. We believe that this and other studies will contribute to the implementation of various safety programs in order to protect public health and road safety.

**Author Contributions:** Conceptualization, A.H.; methodology, A.H.; software, J.K.; validation, S.H., A.H. and J.K.; formal analysis, A.H.; investigation, A.H.; resources, J.K.; data curation, A.H.; writing—original draft preparation, A.H.; writing—review and editing, J.K.; visualization, J.K.; supervision, S.M. and S.H.; project administration, S.H.; funding acquisition, S.M. All authors have read and agreed to the published version of the manuscript.

**Funding:** This work was supported by the Slovak Research and Development Agency under the contract No. APVV-18-0457.

**Institutional Review Board Statement:** Not applicable.

**Informed Consent Statement:** Not applicable.

**Data Availability Statement:** Material and data are available, for example, in CSV format, or on request in another format.

**Conflicts of Interest:** The authors declare no conflict of interest.

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
