# Peer review of "A Study of the Behavior and Responsibility of Slovak Drivers, Especially in Case of Fatigue"

_applsci, doi:10.3390/app11178249_

Round 1
Reviewer 1 Report
Generally, the manuscript is interesting. However, this topic has been widely studied before from different points of view, and I am not sure this paper can provide new useful information. On the other hand, the writing style should be reviewed to correct the multiple errors in the manuscript.
Introduction:
- The introduction is too long and some information is unnecessary or repeated. It feels like the third and fourth paragraphs (lines 63-69 and 70-85, respectively) should be placed just before the first paragraph so that all the information about driving fatigue and safety driving is put together. These three paragraphs, as well as the text containing information about the temperature of the vehicle, should be synthesized.
- Lines 38 and 39 repeat the same idea and sound alike, so please synthesize this part or remove one of the sentences.
- The format of the references in line 66 is different than that in lines 64, 78, or 82. Please check this in the whole manuscript and put them correctly.
- Where does this definition come from? Line 71 lacks a reference.
Materials and methods:
- In line 152, authors said that the questionnaire has been approved, but it seems that it has not been validated… I think this is a limitation, especially considering that, currently, there are already enough validated questionnaires to evaluate drivers’ behavior. Has this questionnaire been used in previous works or in a pilot study? Which criteria did the authors used to create it? Is this questionnaire based on previous literature or other validated questionnaires?
- For me, it is not clear what do you mean when you refer to “young drivers” in the methods. Which age range was considered as “young”? In lines 156-157, authors said that they obtained a sufficient number of questionnaires from young drivers. What number did you consider as “sufficient”? Did you perform a power analysis to check this? If not, authors should include this analysis in this experiment.
- Authors said that the questionnaire was divided into three parts. Which parts? I think this information is not included in table 1.
- In tale 1, there is an error at the end of question 9 (references and an error message). Please remove it.
- In line 169, at the beginning of the paragraph, when authors say “The first included information… The second one concerned…” The first what? Please, complete the sentence.
Results
- Generally, tables and figures captions should provide more detail about the information and data contained on them.
- There are too many tables in the results and, in my opinion, some of them are unnecessary. For example, tables 3, 5, and 7 could be removed, so that results from Chi-square and Mann-Whitney tests would appear only in the text instead.
- Figure 2 is confusing; characters are not distinguishable. Please, improve the quality of this figure or just remove it.
Discussion
- The discussion is poor. The main problem is that authors have just comment their results, but they did not refer to previous literature. As commented before, this topic has been studied and discussed by many authors, and I think their findings would help to better understand and explain the results.
- It seems that the same conclusion is repeated in lines 454-456. Please, remove one of the sentences.
Author Response
Reviewer 1
Generally, the manuscript is interesting. However, this topic has been widely studied before from different points of view, and I am not sure this paper can provide new useful information. On the other hand, the writing style should be reviewed to correct the multiple errors in the manuscript.
Introduction:
- The introduction is too long and some information is unnecessary or repeated. It feels like the third and fourth paragraphs (lines 63-69 and 70-85, respectively) should be placed just before the first paragraph so that all the information about driving fatigue and safety driving is put together. These three paragraphs, as well as the text containing information about the temperature of the vehicle, should be synthesized.
The introduction and each paragraph were synthesized. Lines 63-69; 70-85 were placed to the beginning of the text.
This questionnaire includes a unique combination of questions, and the country in which it took place is also of great benefit, because the countries of Central Europe receive little attention.
A recent study on the occurrence of microsleep in drivers, their responsibility and attitude in the occurrence of this phenomenon was missing in Slovakia but also in the surrounding countries. (i.e., Poland, Czech, Hungary, Austria) It was one of the reasons for starting the research. In addition, we believe, that it is very important to do more road safety research in countries that have traditionally received very little attention.
- Lines 38 and 39 repeat the same idea and sound alike, so please synthesize this part or remove one of the sentences.
We agree, these lines were removed from the manuscript.
- The format of the references in line 66 is different than that in lines 64, 78, or 82. Please check this in the whole manuscript and put them correctly.
The references were modified to the required format throughout the manuscript.
- Where does this definition come from? Line 71 lacks a reference.
A reference in support of this claim has been added.
Materials and methods:
- In line 152, authors said that the questionnaire has been approved, but it seems that it has not been validated… I think this is a limitation, especially considering that, currently, there are already enough validated questionnaires to evaluate drivers’ behavior. Has this questionnaire been used in previous works or in a pilot study? Which criteria did the authors used to create it? Is this questionnaire based on previous literature or other validated questionnaires?
It was a problem to find a similar questionnaire that would focus directly on this issue, especially on responsibility, the thermoneutral temperature in the car interior and so on. We can say that the questionnaire combines unique questions into one compact unit, which is used for our further research. The questionnaire was also designed on the basis of the studies presented in the introduction.
The first step in validating a survey was to establish face validity. Then the experts and people who understand this topic read through a questionnaire. It was stated that questions effectively capture the topic under investigation. The second step was a pilot test of the survey for a group of our intended population. The pilot test was attended by 80 participants. After data collection and feedback from respondents, the data were adjusted and the main components were analysed. The questionnaire was finally modified and revised.
- For me, it is not clear what do you mean when you refer to “young drivers” in the methods. Which age range was considered as “young”? In lines 156-157, authors said that they obtained a sufficient number of questionnaires from young drivers. What number did you consider as “sufficient”? Did you perform a power analysis to check this? If not, authors should include this analysis in this experiment.
We have defined young drivers as drivers under the age of 25. We also used this division on the basis of the mentioned literature and studies. We have misinterpreted a number of young drivers with this statement. We have fixed this mistake, see manuscript. The most numerous sample of respondents were young drivers.
- Authors said that the questionnaire was divided into three parts. Which parts? I think this information is not included in table 1.
The division of the questionnaire was meant more than formal for the authors and readers of the article. It was irrelevant to the final respondent.
- In tale 1, there is an error at the end of question 9 (references and an error message). Please remove it.
This mistake is corrected.
- In line 169, at the beginning of the paragraph, when authors say “The first included information… The second one concerned…” The first what? Please, complete the sentence.
The sentence concerned the division of the questionnaire. This mistake is fixed. See manuscript.
Results
- Generally, tables and figures captions should provide more detail about the information and data contained on them. There are too many tables in the results and, in my opinion, some of them are unnecessary. For example, tables 3, 5, and 7 could be removed, so that results from Chi-square and Mann-Whitney tests would appear only in the text instead.
Unnecessary tables have been removed, see manuscript.
- Figure 2 is confusing; characters are not distinguishable. Please, improve the quality of this figure or just remove it.
Figure 2 has been removed.
Discussion
- The discussion is poor. The main problem is that authors have just comment their results, but they did not refer to previous literature. As commented before, this topic has been studied and discussed by many authors, and I think their findings would help to better understand and explain the results.
Chapter 4, together with the conclusion, has been extended.
- It seems that the same conclusion is repeated in lines 454-456. Please, remove one of the sentences.
Corrected
Reviewer 2 Report
The article addresses a very important issue of the impact of drivers' behavior on road safety. Research related to the behavior of drivers is quite frequent, so it is worth extending the introduction a bit, even by the work of researchers from Slovakia and neighboring countries, such as Poland. The authors focused on several aspects related to driving like age, experience, fatigue, micro sleep, cabin temperature.
What was the reason for choosing these parameters? Have any preliminary analyzes been made, on the basis of which the analyzed parameters have been selected?
In the introduction, the authors should indicate how the research method they used differs from other tests this type undertaken by researchers. They should underline what is new about it.
The article is based on a questionnaire, the content of which is maintained in the text of the 2nd chapter Materials and Methods. However, it would be worthwhile in Chapter 2 to first present (e.g. in the form of a table) the questionnaire itself and later in chapter 3 to present results. The method of presenting the questionnaire used by the authors (chapter 2) makes it difficult to understand its content. In line 166 (chapter 2) was invoked a table 1, containing the results from chapter 3 (line 263). Similarly, in line 184 (rozdział 2) was invoked a table 4 from chapter 3 line 299.
Sections 2.1 and 2.2 briefly discuss the Statistical test methods and the Neural network used. Chapter 3 deals in detail with the results of the questionnaire and additional statistical analyzes. The presented results are very interesting, despite the fact that they are based on subjective opinions of drivers, but they carry a lot of interesting information. The results were clearly presented.
Chapter 4 discusses the obtained results. If it was possible to compare the obtained results of the questionnaire with those obtained by other researchers, it could be an interesting supplement to the present study.
Interesting content could be found in the section Future research directions and limitations, which dispels some doubts about the adopted scope of research and defines the future scope of research. Conclusions are too short and they need to be extended.
The article is very interesting and after some minor corrections, I suggest that it be published.
Additional comments.
In Tables 1-7 and in the text of the paper in decimal values, a comma is used and there should be dots.
In Figure 1 (line 366) descriptions of the axes and the legend should be included.
Descriptions in Table 9 in the "Layer" line should be refilled so that they are consistent with Figure 2.
Author Response
Reviewer 2
The article addresses a very important issue of the impact of drivers' behavior on road safety. Research related to the behavior of drivers is quite frequent, so it is worth extending the introduction a bit, even by the work of researchers from Slovakia and neighboring countries, such as Poland.
As it was noted that the introduction is extensive, this comment was included into the discussion and partly into the conclusion.
The authors focused on several aspects related to driving like age, experience, fatigue, micro sleep, cabin temperature.
What was the reason for choosing these parameters? Have any preliminary analyzes been made, on the basis of which the analyzed parameters have been selected?
This article is part of a wider study of a driver monitoring systems for attention and health monitoring. One of the last articles of the ongoing research can be found at this address: https://www.mdpi.com/1424-8220/21/16/5285
The interior temperature was included in the questionnaire because there is little direct research on the effect of temperature on the driver concentration. We, therefore, needed to collect driver data so that we could perform experimental measurements that are already underway. With questions about responsibility, we wanted to find out the current issue in the country, because data of a similar nature were missing. This information is also used for further comparison and statistics in the future.
The first step in validating a survey was to establish face validity. Then the experts and people who understand this topic read through a questionnaire. It was stated that questions effectively capture the topic under investigation. The second step was a pilot test of the survey for a group of our intended population. The pilot test was attended by 80 participants. After data collection and feedback from respondents, the data were adjusted and the main components were analysed. The questionnaire was finally modified and revised.
In the introduction, the authors should indicate how the research method they used differs from other tests this type undertaken by researchers. They should underline what is new about it.
New in this test is a unique combination of questions, and the country in which it took place is also of great benefit, because the countries of Central Europe receive little attention.
A recent study on the occurrence of microsleep in drivers, their responsibility and at-titude in the occurrence of this phenomenon was missing in Slovakia but also in the surrounding countries. (i.e., Poland, Czech, Hungary, Austria) and it was one of the reasons for starting the research. In addition, we believe, that it is very important to do more road safety research in countries that have traditionally received very little attention.
The article is based on a questionnaire, the content of which is maintained in the text of the 2nd chapter Materials and Methods. However, it would be worthwhile in Chapter 2 to first present (e.g. in the form of a table) the questionnaire itself and later in chapter 3 to present results. The method of presenting the questionnaire used by the authors (chapter 2) makes it difficult to understand its content.
Due to the size of the introductory section and the materials and methods section, we did not want to display the questions of the questionnaire twice. We believe that after the changes, the article is more understandable for the reader.
In line 166 (chapter 2) was invoked a table 1, containing the results from chapter 3 (line 263). Similarly, in line 184 (rozdział 2) was invoked a table 4 from chapter 3 line 299.
Sections 2.1 and 2.2 briefly discuss the Statistical test methods and the Neural network used. Chapter 3 deals in detail with the results of the questionnaire and additional statistical analyzes. The presented results are very interesting, despite the fact that they are based on subjective opinions of drivers, but they carry a lot of interesting information. The results were clearly presented.
Chapter 4 discusses the obtained results. If it was possible to compare the obtained results of the questionnaire with those obtained by other researchers, it could be an interesting supplement to the present study. Interesting content could be found in the section Future research directions and limitations, which dispels some doubts about the adopted scope of research and defines the future scope of research. Conclusions are too short and they need to be extended.
Chapter 4, together with the conclusion, has been extended.
The article is very interesting and after some minor corrections, I suggest that it be published.
Additional comments.
In Tables 1-7 and in the text of the paper in decimal values, a comma is used and there should be dots.
Corrected. See manuscript.
In Figure 1 (line 366) descriptions of the axes and the legend should be included.
Corrected. See manuscript.
Descriptions in Table 9 in the "Layer" line should be refilled so that they are consistent with Figure 2.
Corrected and figure 2 has been deleted due to redundancy.

Round 2
Reviewer 1 Report
The manuscript has been improved following the previous indications. However, there is still a remaining issue to be accomplished before the acceptance of the manuscript: the removed content from tables 3, 5, and 7 should be included in the text, at least the p-value and the statistical value.
Author Response
The manuscript has been improved following the previous indications.
Thank you very much.
However, there is still a remaining issue to be accomplished before the acceptance of the manuscript: the removed content from tables 3, 5, and 7 should be included in the text, at least the p-value and the statistical value.
Key content for understanding the issue has been added.
the p-value parameter from the tables has been added. Confirmation or refutation of the hypothesis was also added.
